# Fungicolous Fungi on Pseudosclerotial Plates and Apothecia of *Hymenoscyphus fraxineus* and Their Biocontrol Potential

**DOI:** 10.3390/microorganisms10112250

**Published:** 2022-11-14

**Authors:** Tadeusz Kowalski, Piotr Bilański

**Affiliations:** Department of Forest Ecosystems Protection, University of Agriculture in Krakow, Al. 29 Listopada 46, 31-425 Krakow, Poland

**Keywords:** ash dieback, antagonism, mycoparasitism, dual cultures, biological control

## Abstract

In the present work, research tasks were carried out in the search for fungi with potential biocontrol possibilities in relation to the ash dieback pathogen, *Hymenoscyphus fraxineus*. In the years 2012–2021, dead petioles of *F. excelsior* and *F. mandshurica* were collected, on which morphological structures of *H. fraxineus* showed unusual symptoms of dying (apothecia) and signs of colonization by other fungi (pseudosclerotial plates). Based on morphological and molecular phylogenetic data, 18 fungal taxa were identified. Thirteen of them belong to Ascomycota: *Clonostachys rosea*, *Cl. solani*, *Cordyceps* sp., *Minimidochium* sp., *Nemania diffusa*, *Fusarium* sp., *Pestalotiopsis* sp., *Trichoderma atroviride*, *T. harzianum*, *T. polysporum*, *T. rodmanii*, *T. tomentosum*, *Trichoderma* sp., and five other taxa are represented by Basidiomycota: *Corticiales* sp., *Cyathus olla*, *Efibula* sp., *Gymnopus* sp. and *Polyporales* sp. In 108 dual cultures in vitro, three different types of interactions were distinguished: (i) physical colony contact (5.6%), (ii) presence of an inhibition zone between the colonies (0.9%), and (iii) copartner overgrowth of *H. fraxineus* colonies and partial or complete replacement of the pathogen (93.5%). In the dual cultures, various morphological deformations of *H. fraxineus* hyphae were observed: the development of apical or intercalary cytoplasmic extrusions, development of internal hyphae of the test fungi in pathogens’ hyphae, the deformation and disruption of significant sections of *H. fraxineus* hyphae via lysis and mycoparasitism, complete desolation of *H. fraxineus* cells and breakdown of hyphae into short fragments, and disappearing of pigment in the affected hyphae of *H. fraxineus*. The inoculation tests performed in vivo or in glass Petrie dishes showed that all the identified taxa were able to lead to pathological changes in *H. fraxineus* apothecia, and the mycelium of some of them completely covered pseudosclerotial plates of *H. fraxineus*. It was emphasized in the discussion that such activity of these fungi in forest stands may contribute to the reduction in the *H. fraxineus* inoculum reservoir.

## 1. Introduction

Ash dieback caused by an alien acomycete *Hymenoscyphus fraxineus* has been the most dangerous *Fraxinus excelsior* disease in most European countries since the 1990s [1,2,3,4]. Trees of all age classes are at risk, but generally young plantations suffer more than older trees [5,6,7]. The disease leads to a significant reduction in the population of *F. excelsior* in forest ecosystems and urban green areas in many European countries [8,9]. In Great Britain, for example, the total economic cost of ash dieback is estimated at £ 14.8 billion [10].

Various silvicultural methods are used to reduce the losses caused by the disease, depending on the severity of damage, local climate, age, and stand type [2,6,11]. The possibilities of reduction in the amount of infectious material [12,13] and methods of eliminating the pathogen from seeds and seedlings were indicated [14]. Some fungicides proved to be effective against *H. fraxineus*. However, they cannot be used on a wider scale in forest stands due to ecological and public health considerations [15]. Despite the measures undertaken, the establishment of new ash plantations in many European countries is not recommended because of the high losses in ash stands [2,16,17].

These are the reasons why alternative disease management products need to be developed, that is, in order to offer environmentally friendly and economically feasible control of this dangerous tree disease. An important element in integrated disease management strategies could be biological control. It involves the use of any living organism to extinguish a certain plant pathogen by means of competition for resources or space, antibiosis, and parasitism [18,19]. Biocontrol agents or microbial antagonists prevent infection or the establishment of the pathogen in the host plant. Thus far, approximately 300 taxa belonging to 113 fungal genera have been identified as biological control agents for plant fungal pathogens [20].

Among the main beneficial fungi with biocontrol capacity are endophytes [21]. Recent studies, conducted mainly in vitro, show that the endophytes in ash leaves and shoots are very diverse and may have a high potential for antagonistic influence toward *H. fraxineus* [22,23,24,25,26]. The usefulness of *Thielvia basicola*, an endophyte in ash petioles, was confirmed in tests with callus cultures and in vitro regenerated ash seedlings [27]. The antagonistic potential of ash endophyte *Hypoxylon rubiginosum*, producing the antifungal beta-tubulin phomopsidin inhibitor, has also been demonstrated [28]. Schlegel et al. [29] found that exudates of ash endophytes, *Boeremia exigua*, *Kretzschmaria deusta*, *Neofabraea alba, Paraconiothyrium* sp., and *Fraxinicola* (*Venturia*) *fraxini* inhibited *H. fraxineus* ascospore germination. Barta et al. [30] observed that the growth of *H. fraxineus* was the most inhibited by four endophytic fungi from ash twigs, *Fusarium lateritium*, *Didymella aliena*, *D. macrostoma*, and *Dothiorella gregaria*. Two of them, *F. lateritium* and *D. aliena*, caused significant inhibition of the length of necroses caused by *H. fraxineus* after inoculation in the same trunks of *F. excelsior* in situ.

After infected ash leaves fall to the bottom of the stand, *H. fraxineus* begins the saprotrophic phase. In this stage, a large role is played by saprotrophs, which may limit the expansion of *H. fraxineus* through the competition for space or antibiosis. Kowalski and Bilański [31] found over 130 fungal taxa in the previous year ash petioles. They showed that the most common competitors for *H. fraxineus* in situ were: *Cyathicula coronata*, *C. fraxinophila*, *Hypoderma rubi*, *Hymenoscyphus caudatus*, *H. scutula*, *Leptosphaeria* sp., *Pyrenopeziza petiolaris* and *Typhula* sp. The highest antibiotic activity against *H. fraxineus* in dual cultures was shown in *Pseudocoleophoma polygonicola* and *Malbranchea* sp. [31].

Among the biocontrol agents, an important group of fungi used in the environmentally sustainable approach to management of plant diseases are mycoparasites [19,32,33]. Mycoparasitism is based on the physical penetration of the parasite into the host hyphae via the development of specific organs such as haustoria, or through its growth inside hyphal cells, and secretion of enzymes or secondary metabolites leading to the degradation of fungal structures followed by nutrient uptake from the host fungus [34,35,36]. The term mycoparasitism applies strictly to those relationships in which one living fungus acts as a nutrient source for another. However, fungicolous fungi have a stable but indeterminate connection with another fungus, though usually presenting a true parasitic relationship is a difficult task [34].

Mycoparasitism can be classified as a four-step process: (a) chemotropic growth of biocontrol fungus mycelium towards the target pathogen, (b) recognition, (c) direct attachment and coiling, and (d) penetration and digestion [35,37]. Mycoparasitic relationships can be biotrophic or necrotrophic. Biotrophic mycoparasites obtain nutrients from living cells without harming the mycohost in a distinct way. The necrotrophic parasite destroys the host cell after or slightly before invasion and utilizes nutrients from the dying or dead host. The invasion is often initiated by coiling of hyphae around the mycohost cells and results in rapid destruction. The necrotrophic mycoparasites are considered to have the best potential as biocontrol agents. Some of them have been used for the commercial control of plant diseases [34,37,38,39,40,41].

The mycoparasites represent different taxonomic units and differ in terms of the extent of the mycohosts. Mycoparasites that can infest a wide range of fungi are classified as generalist, and those that occur on only one or a small number of hosts are referred to as specialist [40]. The most commonly encountered and studied mycoparasitic fungi that infest the broad-spectrum of phytopathogens are mainly *Clonostachys rosea* and *Trichoderma* spp. [20,37]. *Ampelomyces quisqualis* has been considered a mycoparasite that can infect dozens of species of powdery mildew. However, recent studies show that it is genetically diverse and may include several cryptic species [39]. Additionally, *Eudarluca caricis*, which was assumed to be a nonspecific mycoparasite of rust fungi, according to new findings, is not genetically uniform and shows host specificity [42]. *Paraphaeosphaeria* (*Coniothyrium*) *minitans* is a mycoparasite of *Sclerotinia* spp. and other related species. In the fungus *Sclerotinia sclerotiorum* it parasitizes both the hypha and sclerotia [43]. The soilborne oomycete, *Pythium oligandrum* can parasitize on many pathogens such as *Rhizoctonia solani*, *Fusarium oxysporum* and *Phytophthora megasperma* [44]. *Verticillium biguttatum* is a mycoparasite of the dangerous soil-borne fungus *Rhizoctonia solani* and several other plant pathogens [45]. The basidiomycete, *Xenolachne longicornis*, is a known parasite of some ascomycetes of the Helotiaceae family, including *Hymenoscyphus vernus* [46,47]. Thus far, there are no reports of fungicolous or mycoparasitic fungi on ash dieback pathogen, *H. fraxineus*.

The subject of the present work was research on the occurrence of fungicolous and mycoparasitic fungi on ash dieback pathogen, *H. fraxineus*, with the assumption that determining the belonging of the encountered fungi to one of these groups separately will not be easy and at the current stage will have to be considered together. The aims of this study were: (i) identification of fungi occurring on prematurely dead apothecia and on psedosclerotial plates of *H. fraxineus* in situ, (ii) determining the types of in vitro interactions in dual cultures between these species and *H. fraxineus*, and (iii) conducting inoculation tests to evaluate the effect of these fungi on the overgrowth of pseudosclerotial plates and the death of the apothecia of *H. fraxineus* developed in the previous year ash petioles, which would be associated with a reduction in the inoculum of this pathogen.

## 2. Materials and Methods

### 2.1. Sampling and Isolation of Fungi

In the years 2012–2021, during other ash dieback studies, 45 petioles of *F. excelsior* and *F. mandshurica* from the leaf litter were collected (Figure 1 and Table 1), on which morphological structures of *H. fraxineus* (apothecia, pseudosclerotial plates) showed unusual symptoms and signs of other fungi. The apothecia were gray-brown with some deformations, not typical for *H. fraxineus*. However, the pseudosclerotial plates of *H. fraxineus* were overgrown by various forms of mycelium (Figure 2a–o).

Petioles were placed separately in plastic bags and transported to the laboratory, where the symptoms were described in detail and photomicrographs were taken. Subsequently, various attempts were made to isolate fungi colonizing pseudosclerotia and apothecia of *H. fraxineus*. If spore aggregates were present, they were transferred to a drop of sterilized distilled water with a sterile preparatory needle, and then spread on the surface of the agar medium in Petrie dishes. After the hymenium fragments were collected from the apothecium, the procedure was similar. Attempts were also made to obtain colonies of *H. fraxineus* from brown apothecia (from ascospores), according to the method used by Kraj and Kowalski [48]. In the remaining cases, different combinations of disinfection of sample fragments with 96% ethyl alcohol and/or sterile water (3 × 30 s) were used. After drying them on sterile blotting paper, very small pieces were laid out on medium agar. Mycelia of emerging colonies were regularly transferred on MEA in new Petrie dishes and incubated for 3–4 weeks at 20 °C in darkness.

From nine petioles, it was not possible to isolate cultures whose microscopic features of mycelium or spores would match those of the fungi inhabiting *H. fraxineus* morphological structures in situ. Thus, isolates obtained only from 36 samples were included in the analyzes and experiments.

Fungi were grown on malt extract agar [MEA; 20 g l-1 malt extract (Difco; Sparks, MD, USA), 15 g l-1 Difco agar] supplemented with 100 mg l-1 streptomycin sulfate to suppress bacterial growth. Morphological observations, as well as initial morphological identification, were performed as described by Kowalski et al. [49].

The nomenclature of the taxa followed the Index Fungorum (http://www.indexfungorum.org, accessed 1 September 2022). Representative fungal strains were stored on MEA slants at 5 °C in the culture collection of the Department of Forest Ecosystems Protection, University of Agriculture in Kraków.

### 2.2. DNA Extraction, PCR and Sequencing

To verify the morphology-based identification and to identify non-sporulating morphotypes, the nucleotide sequences of the ITS region of the ribosomal RNA (rRNA) gene cluster of representative cultures were determined.

DNA was extracted from 3-week-old fungal cultures using the Genomic Mini AX Plant Kit (A&A Biotechnology, Gdynia, Poland) according to the manufacturer’s instruction. The ITS region of the rRNA gene cluster was amplified for sequencing using primers ITS1-F [50], ITS4 [51] or ITS5 and ITS4 [51]. Polymerase chain reaction (PCR) amplification and sequencing reactions of the isolates were carried out according to the methods described in detail by Bilański and Kowalski [26].

Searches using the BLASTn algorithm were performed in order to find sequences from NCBI GenBank (http://www.ncbi.nlm.nih.gov, accessed 1 September 2022) [52] and were similar to those obtained in the present study. Selected reference sequences of taxa closely related to the isolates collected in this study were obtained from GenBank with the purpose of conducting phylogenetic analysis. All newly obtained sequences in this study were deposited in GenBank with the accession numbers presented in Table 1.

An ITS sequence matches of ≥98% were used to define species boundaries and matches ≥90% were used to define order boundaries [53]. If sequencing of the ITS region of the rRNA gene cluster was ambiguous, the taxonomy was determined by phylogenetic analysis with reference sequences from the NCBI GenBank.

The ITS region of the rRNA gene cluster sequences were aligned using MAFFT v.6 [54], with the E-INS-i option and the remaining default parameters. The BioEdit v.2.7.5 [55] program was used to manually check the correctness of the alignment obtained this way.

Phylogenetic analyses were performed for ITS dataset, using two different methods, i.e., maximum likelihood (ML) and Bayesian inference (BI). The best-fit evolutionary substitution model for the dataset was established for these methods using the corrected Akaike Information Criterion (AICc) in jModelTest v.2.1.10 [56,57]. The best model that met the assumed criteria was the GTR + I + G. ML analysis and was performed in PhyML 3.0 [58] with 1000 replicates of the bootstrap. The BI analysis was carried out in MrBayes v.3.2.6 [59]. The Markov chain Monte Carlo (MCMC) algorithm was run for 10 million generations using the best-fit model. Trees were sampled every 100 generations, resulting in 100,000 trees from both runs and discarding the first 25% as burn-in. The resulting phylogenetic tree (Figure 3) contain all the ITS region of the rRNA gene cluster sequences generated in this study.

### 2.3. Dual Culture Assay

Obtained fungal isolates (Table 1) were screened for their interaction with the colonies of three strains of *H. fraxineus* (HMC 20952, HMC 21508 and Js 9359, further described as Hf1, Hf2, and Hf3, respectively) by in vitro dual culture trials on malt extract agar (one dish per interaction). All of *H. fraxineus* strains originate from the previous year leaf petioles, with developed pseudosclerotial plates, collected in 2017 from the litter in ash stands in southern Poland. They showed different rates of growth on MEA; after three weeks, the average colony radii were: Hf1-3.5, Hf2-3.9, and Hf3-2.8 cm.

A plug of young, actively growing mycelium (8 mm diam.) of a *H. fraxineus* isolate was placed inside the plate, 25 mm from its margin; 7 days later, a similar mycelium plug of the test fungus was placed 40 mm from the *H. fraxineus* plug and 25 mm from the opposite margin of the dish. Two replicates for each *H. fraxineus* strain without other fungi served as control plates. All these plates were incubated in the dark at 20 °C. The interactions between the pathogen and test fungi were assessed after 3 weeks. Any earlier or later (up to 6 weeks) changes in the cultures were also noted. The width of the inhibition zone (mm) was measured along the axis joining the plugs of both fungi.

Three types of pathogen-test fungus interactions were considered, following Bilański and Kowalski [26]: type A, mutual direct contact, when both colonies meet along the contact line, without inhibition zone; type B, with an inhibition zone between colonies; and type C, the test fungus overgrows and covers the colony of *H. fraxineus*. Within type C, fungi that had only overgrown the *H. fraxineus* colonies weakly (<1.5 cm from the edge of the colony at the time of evaluation) have been noted separately. When a test fungus overgrew and covered the *H. fraxineus* colony, its efficacy against *H. fraxineus* was checked using re-isolations (6 inocula taken from each combination). Microscopic observations of *H. fraxineus* mycelium in the impact zone were performed to verify the occurrence of morphological deformations in comparison to the control cultures. This was done for at least one isolate of each fungal taxon.

### 2.4. Inoculation of Ash Petioles

Inoculations were performed in the second half of July, for some isolates in 2019, and for others in 2021. About 250 previous year leaf petioles of *F. excelsior* with pseudosclerotial plates and apothecia of *H. fraxineus* in ash stands in Miechów-Domiarki and Ojców were collected each year (Figure 1). The experiments were performed in vitro and in vivo (Appendix A). In both situations, a set of five petioles, with a total of more than 20 apothecia of *H. fraxineus*, was used to study the interaction with a given fungal species. These petioles were stacked side by side in parallel. In 2019, on each set of petioles, in their basal and apical parts, four pieces of agar (approx. 2 cm × 0.5 cm) overgrown with about 14-day-old mycelium of a given taxon were placed across the petioles. There was one set per tested interaction used. The controls were petioles without fungal inoculum.

Sterile glass Petri dishes (20 cm diam., 2.8 cm height) were used for the in vitro test. A sterile blotting paper was placed at the bottom of these dishes and was wetted with sterile water at the beginning and approximately every 7 days. Then, three glass rods (5 mm diameter) were placed in parallel to each dish on the blotting paper, approx. 5 cm apart from each other. A set of five petioles was placed perpendicularly on these glass rods, so that the petioles would not lay directly on the blotting paper. After inoculation, Petrie dishes were placed in plastic boxes in a room (approx. 18 °C) with diffused daylight.

The in vivo test was carried out on a fenced area in the form of urban greenery (over 400 m^2^). This site was partly (about 30%) moderately shaded by coniferous and deciduous trees and bushes. The ground was mostly covered with grass with extensive patches of moss. Sets of five petioles were put in different, partially shaded places with previously cut down grass. After inoculation, each set was covered with two living, densely needled shoots, about 30 cm long (*Thuja* sp., *Juniperus sabina*).

In 2021, the test was carried out according to the same methodological assumptions, with one difference regarding the inoculum. In order to obtain inoculum, the fungi have been grown for two weeks in darkness at room temperature on MEA. Subsequently, sterile 3-cm-long fragments of ash petioles were placed on the colonies and incubated for two additional weeks. Four such fragments, overgrown by the test fungus, were placed in parallel between the petioles so that they adhered to them as closely as possible.

The inoculated petioles were observed every 5–7 days with the use of a magnifying glass, and the observed changes have been noted. A final evaluation of the condition of the pseudosclerotial plates and apothecia of *H. fraxineus* was done after 17 to 21 days. Mycelium (spores) on pseudosclerotial plates and changes, indicating death or inactivation of apothecia, have been noted. To estimate the intensity of this phenomenon, the following scale was adopted: scarce (<10% of dying apothecia), moderately numerous (11–50%), and very numerous (>50%). The share of a given taxon in the process of colonization of petioles and death of apothecia was confirmed by re-isolation (six fragments for each taxon) and through microscopic examination.

## 3. Results

### 3.1. Fungi Occurring on Pseudosclerotial Plates and Apothecia of H. fraxineus

A total of 36 petioles colonized by *H. fraxineus* with developed pseudosclerotial plates and apothecia were analyzed. They were mostly petioles of *F. excelsior* (31), while a few (5) belonged to *F. mandshurica*. Overall, on eleven petioles, apothecia occurred either singularly or in groups that showed dying symptoms. Both the stipe and receptacle were brown in such apothecia. The disc was sometimes wrinkled and smaller compared to the live neighboring white apothecia (Figure 2a–f and Table 1).

However, on 25 petioles, on pseudosclerotial plates of *H. fraxineus*, various forms of mycelium were present, sometimes with spores (Figure 2g–o and Table 1). In total, 18 fungal taxa that colonized the morphological structures of *H. fraxineus* were identified (Figure 3 and Table 1). Thirteen of them belong to Ascomycota: *Clonostachys rosea* (5 samples), *Cl. solani* (2), *Cordyceps* sp. (3), *Minimidochium* sp. (1), *Nemania diffusa* (1), *Fusarium* sp. (4), *Pestalotiopsis* sp. (2), *Trichoderma atroviride* (3), *T. harzianum* (2), *T. polysporum* (1), *T. rodmanii* (1), *T. tomentosum* (1), and *Trichoderma* sp. 1 (3) (Table 1). Whereas five taxa are represented by Basidiomycota: *Corticiales* sp. (1), *Cyathus olla* (1), *Efibula* sp. (1), *Gymnopus* sp. (1), and *Polyporales* sp. (3) (Figure 3 and Table 1). The currently isolated *Fusarium* sp. differed morphologically and genetically from the endophytic *Fusarium* species (FeC23, FeC44) and saprotrophic *Fusarium* species (FeF132, FeF133) isolated from ash petioles in South Poland (Figure 3).

Four taxa, *Corticiales* sp., *Minimidochium* sp., *T. harzianum* and *T. rodmanii,* were found only on apothecia, while the rest were found either only on pseudosclerotial plates or on both types of substrate (Table 1). On one petiole, next to the *T. polysporum* mycelium, dead apothecium of *H. fraxineus* was present, which was also colonized by this fungus species (Figure 2n).

The apothecia were only clearly overgrown with mycelium when they were colonized by *T. atroviride* and *T. harzianum* (Figure 2d,f), while on the remaining apothecia, the mycelium or spore aggregates were poor (Figure 2a–c,e). On most pseudosclerotial plates, the mycelium was spread (Figure 2g,i–m), less often fine powdery clusters of spores (*Cordyceps* sp.), or in the case of *T. polysporum* and *Trichoderma* sp. 1, the cushion-shaped mycelium with spores (Figure 2n,o) were present. *Efibula* sp. produced distinct mycelial cords (Figure 1h). Conidia in situ or in vitro were produced by: *Cl. rosea*, *Cl. solani*, *Cordyceps* sp., *Fusarium* sp., *Pestalotiopsis* sp., and *Trichoderma* species, while sexual morphs were not observed in any taxon.

### 3.2. Competition Test in Dual Cultures

Overall, the observations were made in 108 dual cultures in vitro (3 *H. fraxineus* strains × 36 fungal isolates). In dual cultures, at the time of evaluation, three different types of interactions were distinguished: (i) physical colony contact, (ii) presence of an inhibition zone between the colonies, and (iii) copartner overgrowth of *H. fraxineus* colonies (Table 2). The frequency of these types varied considerably. The physical contact of the colonies of both partners (type A) was observed only in 5.6% of the dual cultures, in those variants where the copartners were two *Cordyceps* sp. isolates (Table 2, Figure 4a). In the following two weeks, these isolates also slightly overgrew the colonies of *H. fraxineus* and produced conidia on them. An inhibition zone (type B) was only visible in one dual culture, between *H. fraxineus* (Hf1) and *Minimidochium* sp. (Figure 4b and Table 2). Such a zone was present earlier (after 2 weeks) in all three dual cultures with this fungus. However, *Minimidochium* sp. in dual cultures with two *H. fraxineus* strain (Hf2 and Hf3) started to produce a delicate mycelium that outgrew the inhibition zone and started to slightly overgrow the pathogen colony. The development of this mycelium only in a limited zone may indicate chemotropism (Figure 4c and Table 2).

In a third type of interaction (type C), the *H. fraxineus* colony became overgrown by the test fungus. It happened in 101 (93.5%) of the dual cultures (Table 2). The intensity of their mycelium development on the pathogen colony was varied. To a small extent (less than 1.5 cm from the edge of the colony), *H. fraxineus* was overgrown in 26 dual cultures by: *Corticiales* sp., *Cordyceps* sp. Fe446K, *Minimidochium* sp., *Fusarium* sp. and *Pestalotiopsis* sp. (Figure 4d,e and Table 2). Even after the evaluation period, the development of their mycelium was relatively slow. However, the colonies of *H. fraxineus* were covered more intensively by: *Cl. rosea*, *Cl. solani*, *Cy. olla*, *Efibula* sp., *Gymnopus* sp., *Polyporales* sp., *N. diffusa* and *Trichoderma* species (Figure 4f–l and Table 2). Four of them: *Cy. olla*, *Polyporales* sp., *T. atroviride* and *Trichoderma* sp. 1 complete replaced *H. fraxineus* colonies until evaluation, the others did so one–three weeks after evaluation. Mycelia of *Cy. olla*, *Efibula* sp. and *Polyporales* sp. after contact with *H. fraxineus* showed a change in structure; mycelium was cord-like with a fan-shaped pattern (Figure 4g,i,k). After re-cultivation from the overgrowth zone, only the test fungal colonies were detected.

In 32 dual cultures (10 fungal taxa) showing a type C interaction, when the colonies of test fungi approached the colony of *H. fraxineus*, their growth was stopped, a narrow inhibition zone was formed, then this zone became overcrossed and their growth continued (Figure 4e,f and Table 2). In addition, it was observed that, in six dual cultures, *H. fraxineus* produced a narrow blackish zone on the side of the copartner’s interaction (Figure 4i and Table 2). In this zone, the melanized hyphae of *H. fraxineus* showed a black-brown color instead of rusty brown. However, among the test fungi, a wide black-brown zone was observed only in *Cy. olla* colonies (Figure 4h).

In the dual cultures, various morphological deformations of *H. fraxineus* hyphae (compared to the control colonies) were observed (Figure 4m–t and Table 2). The most common modifications included: (i) the development of apical or intercalary cytoplasmic extrusions (Figure 4m); (ii) the lysis of hyphae in the zone of physical contact of the copartners, sometimes leading to the complete destruction of *H. fraxineus* cells and breakdown of hyphae into short fragments (Figure 4o,p); and (iii) the deformation and disruption of significant sections of *H. fraxineus* hyphae via lysis and mycoparasitism. In addition, the production of appressoria by *T. tomentosum* and the haustoria-like structures by *T. harzianum* (Figure 4t) was observed. These pathological changes were particularly visible in the thick and pigmented *H. fraxineus* hyphae. Inside such hyphae, the hyaline hyphae of the mycoparasite were visible (Figure 4m,q–t). The pigment disappeared in the affected hyphae of *H. fraxineus* (Figure 4q–t). It is extremely important that *Cordyceps* sp. (Fe 446K), *Efibula* sp. and *Polyporales* sp. are able to attack melanized hyphae of *H. fraxineus* forming a black zone in dual cultures (Figure 4i,n), because melanized hyphae form, in situ, a pseudosclerotial plate of *H. fraxineus*.

### 3.3. Inoculation Test

Petioles inoculations were performed in vivo and/or in glass Petrie dishes (Figure 5a–t and Appendix A). The test showed that all the fungi used were able to lead to pathological changes in *H. fraxineus* apothecia that were not found in apothecia not affected by fungi.

Fungal taxa colonized the ash petioles in a significantly diverse manner. Fungi such as: *Cl. rosea*, *Cl. solani*, *Cordyceps* sp., *Corticiales* sp., *Fusarium* sp., *Minimidochium* sp. and *Pestalotiopsis* sp. developed rather sparse filamentous or floccose mycelium on the surface of the pseudosclerotium (Figure 5a–t). As soon as such hyphae reached the apothecia, brown spots began to appear on the hymenium (Figure 5a,n). With time, the apothecia turned completely brown and numerous hyphae appeared on some of them, and others, *Cl. rosea*, *Cl. solani* and *Cordyceps* sp., also produced conidiophores and conidia (Figure 5b–d). Some fungi (*Cordyceps* sp., *Fusarium* sp.) developed filamentous hyphae, especially between apothecia, showing some form of chemotropism (Figure 5e,m). Four fungal taxa: *Cy. olla*, *Efibula* sp., *Gymnopus* sp. and *Polyporales* sp., developed abundant mycelium on both petioles and apothecia (Figure 5g–j,o), which led to the rapid death of the apothecia (Figure 5h–j,o).

*Efibula* sp. quickly spread to subsequent petioles due to the production of string-like mycelium (Figure 5i). *Trichoderma* species spread relatively quickly within individual petioles, producing mainly cobweb or threadlike mycelium (Figure 5p–t and Appendix A). Only *T. polysporum* produced cushion-like mycelium on petioles and in the vicinity of apothecia (Figure 5s). *T. harzianum* and *Trichoderma* sp. 1 formed abundant clusters of mycelium on colonized apothecia (Figure 5r,t). Atypical development of new *H. fraxineus* apothecia on petioles inoculated with *T. atroviride* (377E, 664K) and *Cl. solani* was observed. Only stipes mostly developed excessively, reaching dimensions of 3–6 × 1.2–1.8 mm. If an apothecial disc developed, it only reached about 2.5 mm in diameter (Figure 3q).

The rate of colonization of petioles by inoculated fungi influenced the number of apothecia of *H. fraxineus* showing pathological changes. Slowly spreading fungi were only able to colonize a few apothecia (<10%) (Appendix A).

## 4. Discussion

In the life cycle of *H. fraxineus*, an important role is played by the saprotrophic stage, which begins after the leaves fall to the bottom of the stand. On leaf residues, especially petioles and thicker veins, the pathogen produces black pseudosclerotial plates. Only on these pseudosclerotia does *H. fraxineus* produce huge amounts of apothecia with ascospores capable of infecting live ash [1,60,61,62]. Current research has shown that both of these *H. fraxineus* structures can be colonized by different fungi. Such apothecia were characterized, among other things, by the fact that they turned brown prematurely and declined. Assessment of the condition of apothecia must be very careful, as typical white apothecia of *H. fraxineus*, without the participation of other fungi, turn ochraceous to dark brown with age [63]. The research revealed 18 fungal taxa that are able to colonize pseudosclerotia and apothecia of *H. fraxineus*. Only a few of these taxa have so far been found on ash petioles studied in Poland. There were only two species, *Clonostachys rosea* and *Nemania diffusa*, living, symptomless petioles in tree crown and in dead petioles lying at the bottom of the stands [26,31]. This indicates that the pseudosclerotia or apothecia of *H. fraxineus* were colonized predominantly by other fungal species than those colonizing symptomless or dead petioles. Clear differences between these fungi could be found on the basis of the interactions in dual cultures in vitro. Currently, in 93.5% of cases, an overgrowth of *H. fraxineus* colonies by the test fungi was found. In the case of endophytes and saprotrophs from ash petioles, the share of this type of interaction was below 10%, and the most common type of interaction with *H. fraxineus* was the creation of an inhibition zone [26,31]. It can be assumed that the dual-culture assay showed different strategies used by various fungal species to colonize various substrates and to obtain the nutrients that are essential for their survival [64].

The deformations currently observed in the mycelia of *H. fraxineus* strains indicate a destructive effect of the test fungi in dual cultures on the pathogen. This effect is very often proved to be lethal to numerous hyphae of *H. fraxineus*, which was associated with apical or intercalar cytoplasmic extrusions. This phenomenon has been observed many times in fungi and is believed to be the result of increasing pressure within hyphal cells affected by toxic organic compounds [26,65]. Other phenomena observed in *H. fraxineus* mycelium, such as local lysis and hyphae fragmentation, loss of cytoplasm and cell walls, development of appresorium-like and haustoria-like structures, development of test fungus hyphae closely adjacent to the pathogen’s hyphae and their development inside pathogen hyphae are characteristic for mycoparasites [35,66,67]. These changes led to such a state that re-isolations of fungi from covered *H. fraxineus* colonies indicated the successful replacement of *H. fraxineus* by the numerous test fungi. The results of the in-dual cultures tests turned out to be largely consistent with the results of the inoculation test performed.

All isolates used in test inoculations were able to make destructive changes in single or multiple apothecia of *H. fraxineus*, leading to their premature death. The observed differences in the development of mycelium on the pseudosclerotial plate on the tested petioles compared to the petioles sampled in situ can be explained by various external conditions, especially temperature and humidity, which largely influence the development of mycelium [65]. Petioles were collected in situ at different times of year, from May to October, and it is not known how long they were colonized by fungi. In the experiment, the period of fungal development after inoculation was limited to a few weeks.

The species of fungi found in the present research differ significantly in terms of lifestyle and ecological role in forest ecosystems, agriculture, or horticulture. The findings of some of them as factors that may affect the reduction of *H. fraxineus* inoculum puts their significance in a completely new light. The best-known biocontrol agents are species of the genus *Clonostachys* and *Trichoderma*.

*Cl. rosea* strains are widely distributed all over the world. They appeared most commonly in soil and different plants organs, such as roots, leaves and flowers [37]. *Cl. rosea* is a destructive parasite against several plant pathogens affecting different crops and forest species [37,68]. For example, it killed spores and vegetative cells of *Ceratocystis fimbriata* [69], parasites on *F. circinatum* hyphae [68], *Rhizoctonia solani* hyphae and sclerotia [70,71] and the *Crinipellis roreri* pseudostroma, a dangerous causal agent of frosty pod rot on *Theobroma gileri* and *T. cacao* [72,73]. *Cl. rosea* can cause decomposition of fungal cell walls due to the ability to produce appropriate enzymes [37]. In the interaction with fungal pathogens, secondary metabolite production often plays a significant role as well [37,40,64]. Several studies have shown that one possible way in which *Cl. rosea* can reduce the disease incidence is by induced systemic resistance [74]. This has recently been shown for *P. radiata*-*F. circinatum* pathosystem [68]. While *Cl. rosea* represents a huge resource with many possibilities, experience in applying of *Cl. rosea* is lacking and requires more research [37].

Fungal species from the genus *Trichoderma* (sexual morph *Hypocrea*) are widely distributed in nature, especially in soil [75]. According to Thambugala et al. [20], *Trichoderma* is the fungal genus with the greatest biocontrol potential. Twenty-five *Trichoderma* species have been used in this capacity against several phytopathogens, such as *Botrytis cinerea*, *Fusarium* spp., *Pythium* spp., *Rhizoctonia solani*, *Sclerotium rolfsii*, and *Sclerotinia sclerotiorum* [20,67,76]. The main biocontrol strategies developed by *Trichoderma* against fungal pathogens are mycoparasitism, competition, antibiosis and systemic induced resistance [67,76]. Lytic enzymes synthesized by *Trichoderma* species, particularly chitinases, β-1,3-glucanases, and proteases are believed to be responsible for their mycoparasitic activity, leading to the degradation of the cell walls of fungal pathogens. Moreover, numerous *Trichoderma* species synthesize a large number of different secondary metabolites that are involved in mycoparasitic action [76,77,78]. *Trichoderma*-based products are marketed worldwide and used for crop protection against numerous disease-causing agents. The living fungal spores are applied in a variety of preparations used for spraying leaves, covering wounds after pruning, treating seeds and propagating material, or by introducing them into the soil before sowing or transplanting plants [40].

The present results support the argument that *H. fraxineus* may also be included in the fungal pathogens that can be suppressed by numerous *Trichoderma* species. This is indicated by both the dual cultures assay and the inoculations test. The mycelium of *Trichoderma* spp. was found on ash petioles in situ, mainly in the period from May to August, i.e., before or during the fructification of *H. fraxineus*, which should be assessed as a positive phenomenon. The appearance of *Trichoderma* in the final stage of petioles degradation would be of less importance for the reduction of the pathogen’s inoculum.

*Minimidochium* species are known mainly as saprotrophic fungi that colonize leaf litter or as endophytes [79,80]. Some species can produce extracellular laccases, which are responsible for the breakdown of lignin in white rot fungi [81]. The observed antibiotic activity of *Minimidochium* sp., lysis of hyphal wall and the possible effect on the death of *H. fraxineus* apothecia are a new aspect for this genus. Studies on the isolate *Minimidochium* sp. Jp49, strongly antagonistic to *H. fraxineus*, showed that it did not damage callus cultures and seedlings of *F. excelsior* and *F. pennsylvanica* [27].

*Fusarium* species exhibit diverse lifestyles. They can occur as saprotrophs, endophytes or plant pathogens, causing serious damage to crop and forest plants. Torbati et al. [82] also found a fungicolous *Fusarium* species associated with smut fungi. This genus also includes mycoparasitic species, which for a long time have been destroying known pathogens such as *Botrytis cinerea*, *Cercospora* spp. and *Rhizoctonia solani* [83,84,85,86]. The current results justify including *Fusarium* sp. to colonize the structure of *H. fraxinus* in the group of mycotrophic fungi. It is a species that is taxonomically different from those found in Poland as endophytes or saprotrophs on ash petioles. Its virulence has not been known. If found to be non-pathogenic, it could be considered as an interesting biocontrol agent against *H. fraxineus*.

*Cordyceps* species (Cordycipitaceae) are known primarily as important entomopathogenic fungi with a worldwide distribution. Numerous studies show that entomopathogenic fungi play very diverse functions in nature. They can act as endophytes, antagonists to plant pathogens or plant growth stimulants. Great research interest currently lies in the possibility of using entomopathogenic fungi as endophytes for the biological control of fungal diseases as well as insect pests [87,88]. In the course of the current research, it could be observed that, when *Cordyceps* sp. reached *H. fraxineus* structures, it proliferated and formed the next generation of spores, thus negatively affecting the pathogen. In dual cultures, despite limited expansion on *H. fraxineus* mycelium, some *Cordyceps* sp. isolates were capable of damaging *H. fraxineus* hyphae, which should be considered as a new aspect in the ecological role of this fungus.

*Pestalotiopsis* species show different modes of life. Among them there are pathogens that cause diseases in a variety of plants, endophytes and saprotrophs in leaves, bark and twigs. Moreover, species of *Pestalotiopsis* have been found to produce very numerous secondary metabolites that may be applied in medicine, agriculture and industry [89,90]. Only in recent years, *Pestalotiopsis* sp. was detected as a mycoparasite of the plant pathogen *Aecidium wenshanense* [91]. This study reveals a large difference between the mechanism of mycoparasitism of *Pestalotiopsis* and that of *Trichoderma* [91]. In *Pestalotiopsis* sp., the number of mycoparasitism related β-1,3-glucanases was greater than that in *Trichoderma*. Recent studies have shown that another fungicolous *Pestalotiopsis* species produces three new sesquiterpenoids related to the caryophyllene-derived punctaporonins [92]. The obtained results suggest that the Polish strain of *Pestalotiopsis* exhibits mycotrophic activity in relation to *H. fraxineus*.

*Nemania diffusa*, like *Cl. rosea*, was found in ash petioles, both as endophyte and saprotroph [26,31]. It cannot be ruled out that its mycelium colonizing petioles in the saprotrophic stage can grow on pseudosclerotial plates of *H. fraxineus* without the need for external infection. The results show that its positive role does not have to only be limited to competition for space inside petioles. *N. diffusa* produces enzymes such as phenoloxidases, peroxidase, acid phosphatase and laccase, which have the ability to neutralize fungitoxic compounds and take part in the decomposition of lignocellulosic substrates [93]. There is also a possibility that they can allow for the destruction of fungal cell walls, as observed in dual cultures in vitro.

Mycoparasitism by basidiomycetous fungi has been known for a long time [94]. The observed interactions in dual cultures and on inoculated petioles seem to confirm such possibilities of the test fungi in relation to *H. fraxineus*. Their antagonistic activity is probably favored by the relatively quick production of mycelium, which abundantly covers ash petioles. They are ecologically important fungi, because they produce enzymes to degrade lignocellulosic material and they may be used as an inoculant to accelerate plant residue decomposition. Identified Basidiomycota species show significant differences in terms of the occurrence and lifestyle. For example, *Efibula* is a genus characterized by resupinate basidiomata classified currently in the family Irpicaceae. They are typically found growing on the underside of dead trees, causing a white rot. *E. tuberculata* is a plant pathogen infecting plane trees [95]. In Brazil a representative of this genus was found to be a frequent endophyte in petioles of the medicinal plant *Vochysia divergens* [90]. *Gymnopus* species are cap-forming fungi that grow on the ground or on dead logs and can create mycorrhiza. Species of this genus belong to the most common endophytes in Orchidaceae [80]. Unlike *Collybia* or *Psathyrella*, there are no known mycotrophic species among *Gymnopus* [96]. *Cyathus olla*, a bird’s nest fungus, is the most abundant species of *Cyathus* found in Europe, and has been reported in other regions of the world. It is a saprotroph, considered to be an aggressive colonizer due to the formation of mycelial cords that invade the substrate. It also spreads through the peridioles and basidiospores. In the last two decades, the role of *Cy. olla* in the biological control of stubble-borne diseases of canola (*Brassica napus*, *B. rapa*) has been studied [97,98]. This species could play a similar role for ash dieback by reducing the inoculum reservoir of *H. fraxineus*.

When evaluating the results of the inoculations test, it should be considered that the inoculum was added only locally on the petioles, so the fungi needed some time before they came into contact with *H. fraxineus* apothecia. Application of spores or fragmented mycelium as a spray directly on the apothecia and along the entire petioles would probably have a more damaging effect on *H. fraxineus*. Observations during the experiment have been conducted for less than a month. However, the impact on the apothecia of *H. fraxineus* in nature may be much greater, as new generations of apothecia forming during the growing season may be invaded by the inoculum formed in large numbers on the originally attacked apothecia. Besides, *H. fraxineus* persists for several years on the petioles, which increases the overall amount of litter inoculum [61]. Mycoparastic fungi may reduce this persistence as they can grow from old petioles to freshly fallen leaves into litter. This may increase the importance of such fungi.

From the fungal taxa briefly presented above, it appears that forest stands (soil, leaf litter or dead wood) are their natural environment. If we were to develop an appropriate methodology and introduce the inoculum of some of these fungi as biocontrol agents, it would not involve the risk of introducing foreign organisms into the environment. Some of these taxa, however, would require further detailed identification and assessment of virulence. The conducted experiments did not provide a final solution, but presented the fungal taxa, within which it is possible to search for an optimal organism for the biocontrol of ash dieback pathogen, *H. fraxineus*. It can be assumed that there would be different applicability of the identified biocontrol fungi against *H. fraxineus*. In numerous places in forests and in larger ash groups in urban green areas, where leaf removal proposed by Noble et al. [13] is impossible, biocontrol fungi could be used to reduce the inoculum reservoir of *H. fraxineus* in the litter. Basidiomycetous species appear to be particularly promising for this purpose [94,95,96,97]. It is possible that the mass colonization of ash petioles in a given year will result in the colonization of freshly fallen leaves in the following years. This would lead to a long-term inoculum reduction. The identified biocontrol fungi could also be used to prevent infections at the base of the stems of especially valuable trees, which often opens the way for subsequent attacks by *Armillaria* species [99,100]. For this purpose, mycoparasitic fungi belonging to Ascomycota could be more useful [20,37].

## 5. Conclusions

Use of beneficial microorganisms in biological control could become an important factor in integrated disease management strategies with regard to ash dieback. Current studies have shown that certain fungi are able to colonize apothecia and pseudosclerotial plates of ash dieback pathogen, *H. fraxineus*, contributing to the reduction of its inoculum. However, this phenomenon occurs relatively rarely in natural conditions. Morphological structures of *H. fraxineus* can be colonized by fungi such as *Clonostachys rosea* and *Trichoderma* species, which are known mycoparasites in the broad spectrum of mycohosts, as well as by more specialized ascomycetous fungi, such as *Cordyceps*, *Fusarium*, *Minimidochium* and *Pestalotiopsis*. An important role should be attributed to the presently found fungi belonging to Basidiomycota: *Corticiales* sp., *Cyathus olla*, *Efibula* sp., *Gymnopus* sp., and *Polyporales* sp., which grow intensively on ash petioles and synthesize enzymes that enable them to intensively decompose lignocellulosic substrates. The conducted research provides indications concerning the potential of many of the identified species of fungi for their use against *H. fraxineus*. A very positive aspect is the fact that these are fungi that occur naturally in forests, soil, leaf litter or dead wood.

## Figures and Tables

**Figure 1 microorganisms-10-02250-f001:**
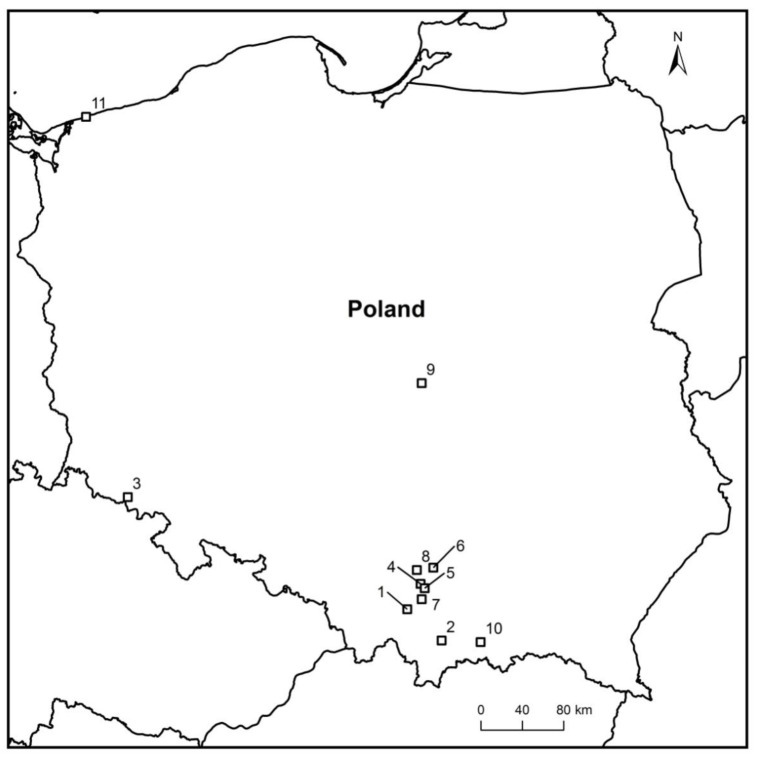
Locations of sampling sites in Poland: 1—Brody, 2—Gorce Konina, 3—Kowary, 4—Kraków Młynówka, 5—Kraków Zakrzówek, 6—Miechów Domiarki, 7—Myślenice, 8—Ojców, 9—Rogów, 10—Stary Sącz, and 11—Trzęsacz.

**Figure 2 microorganisms-10-02250-f002:**
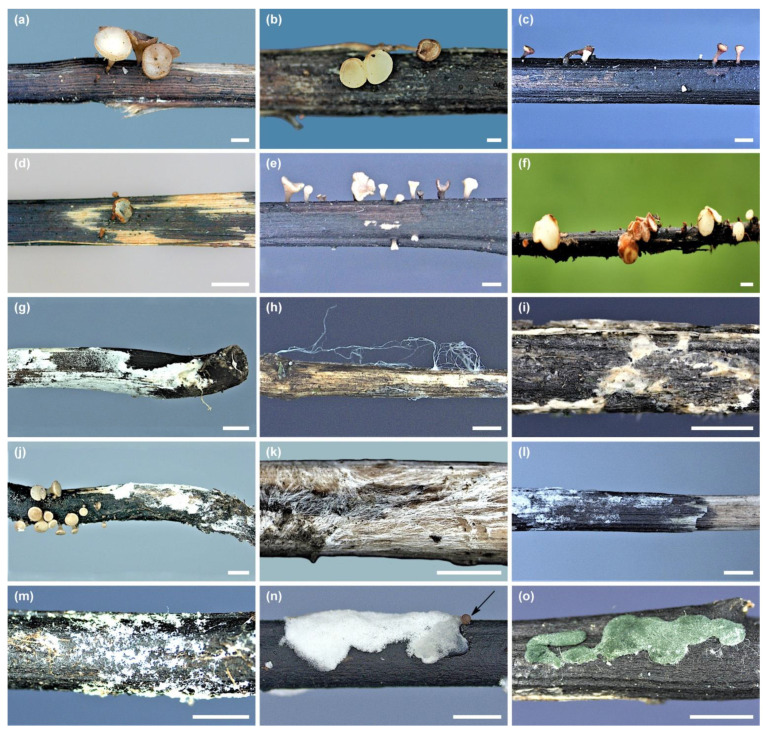
Previous year ash petioles with developed in situ apothecia and pseudosclerotial plates of *Hymenoscyphus fraxineus*: (**a**–**f**) apothecia with degrading symptoms colonized by: (**a**) *Clonostachys rosea* Fe886F, (**b**) *Minimidochium* sp.1 Fe173F, (**c**) *Fusarium* sp. 1 Fm409E, (**d**) *Trichoderma atroviride* Fe377E, (**e**) *Trichoderma harzianum* Fm506K (**f**) *Trichoderma harzianum* Fe412E; (**g**–**o**) pseudosclerotial plate covered with mycelium of: (**g**) *Cyathus olla* Fe141F, (**h**) *Efibula* sp. 1 Fe287F, (**i**) *Gymnopus* sp. 1 Fe1035F, (**j**) *Polyporales* sp. 1 Fe302F, (**k**) *Polyporales* sp. 1 Fe1011F, (**l**) *Pestalotiopsis* sp. 1 Fe447K, (**m**) *Trichoderma atroviride* Fe57K, (**n**) *Trichoderma polysporum* Fe166F—mycelium and dead apothecium (arrow), and (**o**) *Trichoderma* sp. 1 Fe462K; scale bar = 2 mm.

**Figure 3 microorganisms-10-02250-f003:**
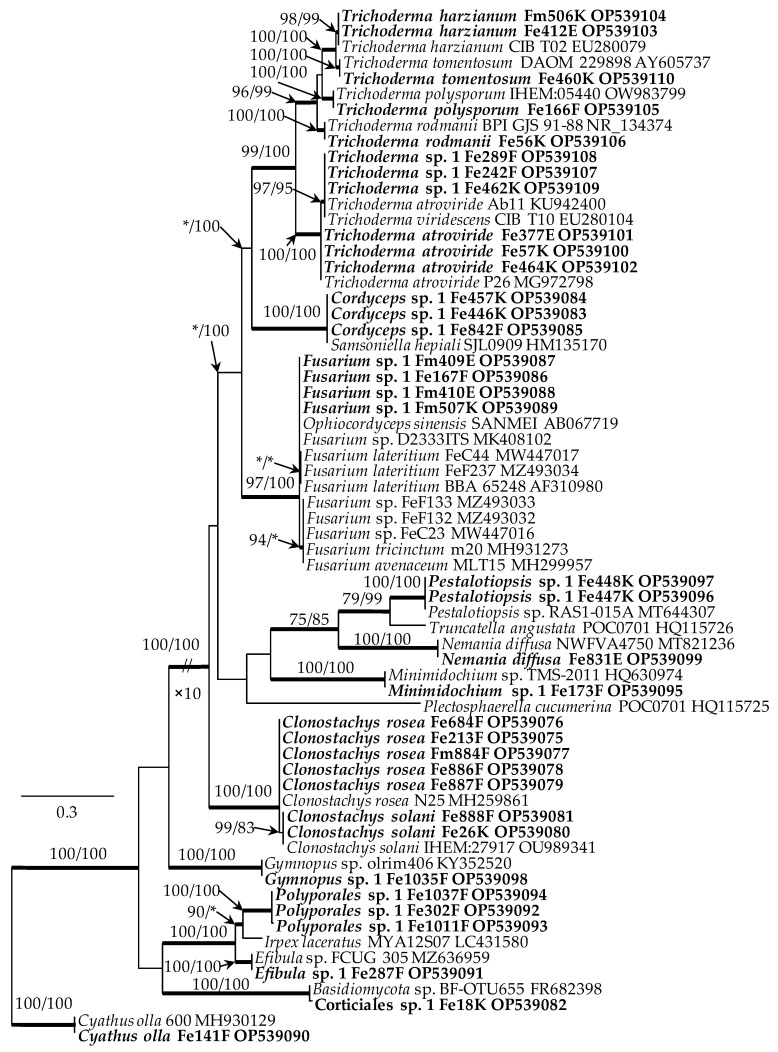
Phylogram obtained from Maximum Likelihood (ML) analyses of the ITS data for the isolated fungal taxa. Sequences obtained during this study are presented in bold type. The Bootstrap values ≥ 75% for ML and Maximum Parsimony (MP) analyses are presented at nodes as follows: ML/MP. Bold branches indicate posterior probabilities values ≥ 0.95 obtained from Bayesian Inference (BI) analyses. * Bootstrap values < 75%. The scale bar corresponds to expected number of substitutions per nucleotide site. *Cyathus olla* was used as the outgroup taxon.

**Figure 4 microorganisms-10-02250-f004:**
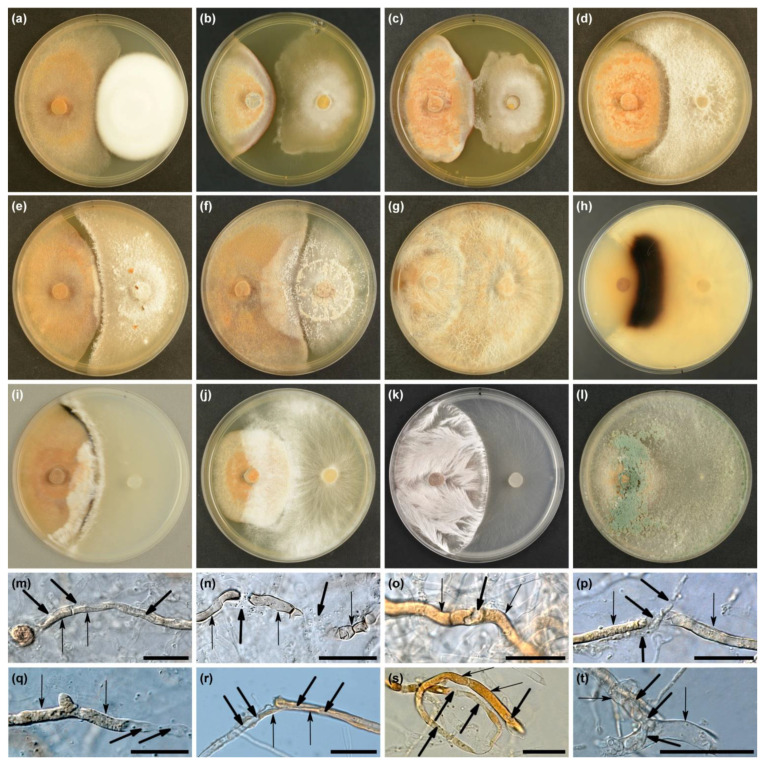
Various types of interactions observed in dual cultures (**a**–**l**) and morphological deformations (**m**–**t**) (MEA, 3 weeks, 20 °C), *Hymenoscyphus fraxineus* (either Hf1, Hf2 or Hf3) from the left, test fungus from the right: (**a**) type A interaction, *Cordyceps* sp. 1 Fe842F, (**b**) type B interaction, *Minimidochium* sp. 1 (Hf1, Fe173F), (**c**–**l**) type C interaction: (**c**) *Minimidochium* sp. 1 (Hf3, Fe173F), (**d**) *Fusarium* sp. 1 Fe167F, (**e**) *Pestalotiopsis* sp. 1 Fe447K, (**f**) *Clonostachys solani* Fe26K, (**g**,**h**) *Cyathus olla* Fe141F (averse and reverse–dark zone), (**i**) *Efibula* sp. 1 Fe287F, (**j**) *Gymnopus* sp. 1 Fe1035F, (**k**) *Polyporales* sp. 1 Fe1037F, (**l**) *Trichoderma atroviride* Fe377E, (**m**–**t**) morphological deformations in *H. fraxineus* hyphae caused by test fungi (thin arrow–*H. fraxineus*, thick arrow–test fungus): (**m**) *Trichoderma rodmanii* Fe56K–cytoplasmic extrusion, internal hypha, (**n**) *Efibula* sp. 1 Fe287F–the deformation and disruption of significant sections of *H. fraxineus* hyphae from the melanized zone, (**o**) *Clonostachys solani* Fe888F–internal hypha, (**p**) *Cordyceps* sp. 1 Fe446K–local lysis and disintegration of pigmented *H. fraxineus* hypha, (**q**) *Corticiales* sp. 1 Fe18K–lysis and disintegration of pigmented *H. fraxineus* hypha, disappearance of the pigment, (**r**) *Fusarium* sp. 1 Fe410E–internal hypha, disintegration of pigmented *H. fraxineus* hypha, disappearance of the pigment, (**s**) *Nemania diffusa* Fe831E–internal hyphae, lysis and deformations of pigmented *H. fraxineus* hypha, disappearance of the pigment, and (**t**) *Trichoderma harzianum* Fe412E–disappearance of the pigment, the haustoria-like structures; scale bar (**m**–**t**) = 25 µm.

**Figure 5 microorganisms-10-02250-f005:**
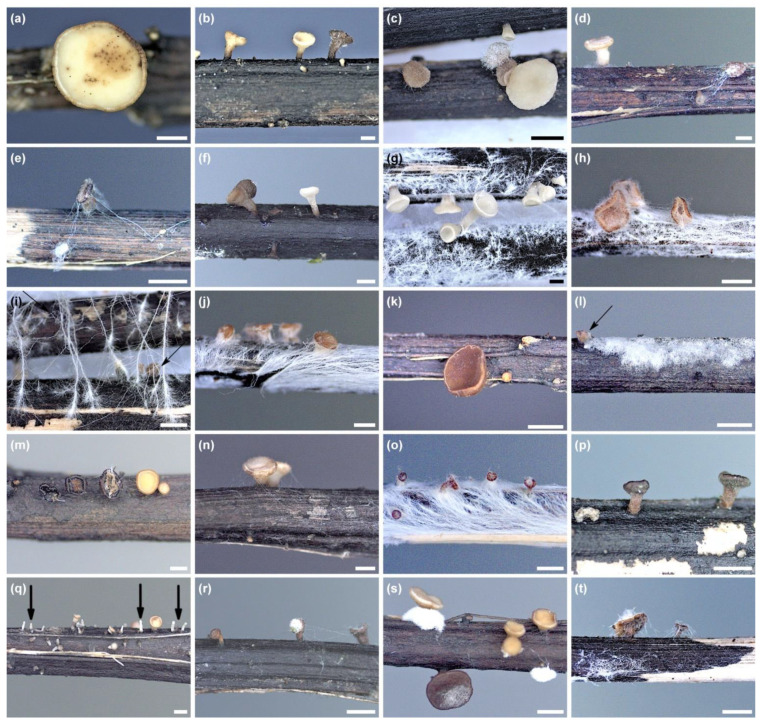
Symptoms of colonization of pseudosclerotial plates and apothecia of *H. fraxineus* for up to three weeks after inoculation of ash petioles with test fungi: (**a**) *Clonostachys rosea* Fe213F, (**b**) *Clonostachys rosea* Fe886F, (**c**) *Clonostachys solani* Fe26K, (**d**) *Cordyceps* sp. 1 Fe446K, (**e**) *Cordyceps* sp. 1 Fe842F, (**f**) *Corticiales* sp. 1 Fe18K, (**g**) *Cyathus olla* Fe141F, (**h**) *Cyathus olla* Fe141F, (**i**) *Efibula* sp. 1 Fe287F–arrows show dead apothecia, (**j**) *Gymnopus* sp. 1 Fe1035F, (**k**) *Minimidochium* sp. 1 Fe173F, (**l**) *Nemania diffusa* Fe831E–arrow shows dead apothecium, (**m**) *Fusarium* sp. 1 Fm409E, (**n**) *Pestalotiopsi* sp. 1 Fe447K, (**o**) *Polyporales* sp. 1 Fe1011F, (**p**) *Trichoderma atroviride* Fe57K, (**q**) *Trichoderma atroviride* Fe377K–arrows show undeveloped and deformed apothecia, (**r**) *Trichoderma harzianum* Fe412E, (**s**) *Trichoderma polysporum* Fe166F, and (**t**) *Trichoderma* sp. 1 Fe462K; scale bar = 2 mm.

**Table 1 microorganisms-10-02250-t001:** Fungi isolated from apothecia and pseudosclerotial plates of *Hymenoscyphus fraxineus* with symptoms of colonization by other fungi. Sequences were deposited in GenBank. Reference sequences from GenBank were obtained using BLAST.

Fungi	Strain Number	Sub. ^1^	Host ^2^	Locality	Collection Date	ITS Accession	BLAST	Identities %
*Clonostachys rosea*	Fe213F	Pp	Fex	Ojców	17 October 2017	OP539075	MH259861	99.83
	Fe684F	Pp	Fex	Miechów Domiarki	12 September 2018	OP539076	MH259861	99.83
	Fm884F	Ap	Fm	Rogów	6 August 2016	OP539077	MH259861	99.83
	Fe886F	Ap	Fex	Stary Sącz	10 July 2014	OP539078	MH259861	99.83
	Fe887F	Ap	Fex	Kowary	6 August 2016	OP539079	MH259861	99.83
*Clonostachys solani*	Fe26K	Pp	Fex	Ojców	26 September 2019	OP539080	OU989341	100
	Fe888F	Pp	Fex	Brody	10 October 2018	OP539081	OU989341	100
*Corticiales* sp. 1	Fe18K	Ap	Fex	Kowary	6 August 2016	OP539082	FR682398	99.54
*Cordyceps* sp. 1	Fe446K	Pp	Fex	Miechów Domiarki	28 August 2018	OP539083	HQ115725	100
	Fe457K	Pp	Fex	Miechów Domiarki	4 October 2013	OP539084	HQ115726	100
	Fe842F	Pp	Fex	Myślenice	5 June 2019	OP539085	HM135170	100
*Cyathus olla*	Fe141F	Pp	Fex	Miechów Domiarki	8 June 2017	OP539090	MH930129	99.87
*Efibula* sp. 1	Fe287F	Pp	Fex	Myślenice	1 December 2017	OP539091	MZ636959	99.83
*Fusarium* sp. 1	Fe167F	Pp	Fex	Brody	10 June 2017	OP539086	AB067719	100
	Fm409E	Ap	Fm	Rogów	12 July 2016	OP539087	MK408102	100
	Fm410E	Ap	Fm	Rogów	12 July 2016	OP539088	MK408102	100
	Fm507K	Ap	Fm	Rogów	12 July 2016	OP539089	MK408102	100
*Gymnopus* sp. 1	Fe1035F	Pp	Fex	Myślenice	20 August 2019	OP539098	KY352520	100
*Minimidochium* sp. 1	Fe173F	Ap	Fex	Kraków Zakrzówek	18 July 2017	OP539095	HQ630974	100
*Nemania diffusa*	Fe831E	Pp	Fex	Kraków Młynówka	12 May 2017	OP539099	MT821236	100
*Pestalotiopsis* sp. 1	Fe447K	Pp	Fex	Ojców	07 May 2021	OP539096	MT644307	100
	Fe448K	Pp	Fex	Ojców	07 May 2021	OP539097	MT644307	100
*Polyporales* sp. 1	Fe302F	Pp	Fex	Miechów Domiarki	20 October 2017	OP539092	LC431580	91.95
	Fe1011F	Pp	Fex	Myślenice	2 August 2019	OP539093	LC431580	91.95
	Fe1037F	Pp	Fex	Myślenice	2 August 2019	OP539094	LC431580	91.95
*Trichoderma atroviride*	Fe57K	Pp	Fex	Brody	10 May 2017	OP539100	MG972798	100
	Fe377E	Pp	Fex	Ojców	4 May 2016	OP539101	MG972798	100
	Fe464K	Pp	Fex	Ojców	10 May 2017	OP539102	MG972798	100
*Trichoderma harzianum*	Fe412E	Ap	Fex	Gorce Konina	27 August 2013	OP539103	EU280079	100
	Fm506K	Ap	Fm	Rogów	12 July 2016	OP539104	EU280079	100
*Trichoderma polysporum*	Fe166F	Pp	Fex	Kraków Młynówka	17 July 2016	OP539105	OW983799	99.84
*Trichoderma rodmanii*	Fe56K	Ap	Fex	Trzęsacz	18 August 2012	OP539106	NR_134374	100
*Trichoderma tomentosum*	Fe460K	Pp	Fex	Kraków Młynówka	17 July 2016	OP539110	AY605737	100
*Trichoderma* sp. 1	Fe242F	Pp	Fex	Miechów Domiarki	5 May 2017	OP539107	KU942400	100
	Fe289F	Pp	Fex	Miechów Domiarki	20 November 2017	OP539108	KU942400	100
	Fe462K	Pp	Fex	Myślenice	26 July 2019	OP539109	EU280104	100

^1^ Substrate: Ap—Apothecium with degrading symptoms, Pp—pseudosclerotial plate overgrown with hyphae or with clusters of conidiophores and conidia. ^2^ Host: Fex—*Fraxinus excelsior*, Fm—*Fraxinus mandshurica*.

**Table 2 microorganisms-10-02250-t002:** Types of interactions of *Hymenoscyphus fraxineus* with test fungi in dual cultures in vitro and morphophysiological deformations in the hyphae of *H. fraxineus*.

Fungal Species	Strain Number	*H. fraxineus* Strains Interaction Type ^1^			Symptoms of Mycoparasitism, Deformations in *H. fraxineus* Hyphae ^2^		
		Hf1	Hf2	Hf3	ME	MH	ML
*Clonostachys rosea*	Fe213F	C	C	Cs (B)	+	+	+
	Fe684F	C	C	C			
	Fm884F	C (B)	C	C			
	Fe886F	C	C	C			
	Fe887F	C	C	C			
*Clonostachys solani*	Fe26K	C	C	C	+	+	+
	Fe888F	C	C	C			
*Corticiales* sp. 1	Fe18K	Cs	Cs	Cs (B)	+	+	+
*Cordyceps* sp. 1	Fe446K	Cs	Cs; Mz	Cs	+	+	+
	Fe457K	A	A	A	–	–	–
	Fe842F	A	A	A			
*Cyathus olla*	Fe141F	C	C	C	+	+	+
*Efibula* sp. 1	Fe287F	C (B); Mz	C (B); Mz	C (B)	+	+	+
*Fusarium* sp. 1	Fe167F	Cs	Cs	Cs	+	+	+
	Fm409E	Cs	Cs (B)	Cs (B)			
	Fm410E	Cs	Cs (B)	Cs (B)			
	Fm507K	Cs	Cs	Cs			
*Gymnopus* sp. 1	Fe1035F	C	C	C	+	+	+
*Minimidochium* sp. 1	Fe173F	B	Cs (B)	Cs (B)	+	–	+
*Nemania diffusa*	Fe831E	C	C	C (B)	+	+	+
*Pestalotiopsis* sp. 1	Fe447K	Cs (B)	Cs (B)	Cs (B)	+	+	+
	Fe448K	Cs (B)	Cs (B)	Cs (B)			
*Polyporales* sp. 1	Fe302F	C (B)	C (B); Mz	C (B)	+	+	+
	Fe1011F	C (B)	C (B)	C (B)			
	Fe1037F	C (B)	C (B); Mz	C (B); Mz			
*Trichoderma atroviride*	Fe57K	C	C	C	+	+	+
	Fe377E	C	C	C			
	Fe464K	C	C (B)	C			
*Trichoderma harzianum*	Fe412E	C	C	C	+	+	+
	Fm506K	C	C	C			
*Trichoderma polysporum*	Fe166F	C	C	C	+	+	–
*Trichoderma rodmanii*	Fe56K	C (B)	C (B)	C (B)	+	+	+
*Trichoderma tomentosum*	Fe460K	C	C	C	+	+	+
*Trichoderma* sp. 1	Fe242F	C	C	C	+	+	–
	Fe289F	C	C	C			
	Fe462K	C	C	C			

^1^ A—physical contact of mycelia; B—inhibition zone; C—overgrowth of *H. fraxineus* colony by test fungus; Cs—slow overgrowth, up to 1.5 cm from the edge of the *H. fraxineus* colony; C (B)—test fungus overgrew *H. fraxineus* colony but there was a very narrow inhibition zone; Mz—melanized blackish zone at the edge of *H. fraxineus* colony. ^2^ ME—apical and intercalar cytoplasmic extrusion; MH—internal hyphae of mycoparasite in hyphae of *H. fraxineus*, the deformation and disruption of significant sections of hyphae; ML—*H. fraxineus*: lysis of hyphae, desolation of hyphal cells, breakdown of hyphae into short fragments, the disappearance of the pigment.

## Data Availability

The data presented in this study are available in Appendix A.

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
