# Peer review of "Fungicolous Fungi on Pseudosclerotial Plates and Apothecia of Hymenoscyphus fraxineus and Their Biocontrol Potential"

_microorganisms, 2022, doi:10.3390/microorganisms10112250_

Round 1

Reviewer 1 Report

Manuscript 1973780 submitted to Microorganisms by Kowalski & Bilanski explores the interaction of fungi found on the ash dieback pathogen pseudosclerotium in the forest floor (Hymenoscyphus fraxineus) with the pathogen and in particular, the role these fungi could have in reducing H. fraxineus presence on the rachis. Ash dieback is a major forest health problem in Europe. H. fraxineus inoculum dynamic strongly depends on the amount of pseudosclerotial rachises in the litter and thus a better understanding of how fungi may affect this potential is of considerable interest. The manuscript represent a nice and very comprehensive advance on the subject. The sampling, on pseudosclerotial rachises/apothecia from the litter that present symptoms, the extensive caracterisation and testing both in Petri dishes and on rachises either in controlled conditions or in the forest litter makes the work very robust and it will without doubt represent a milestone on the subject. In particular, the very nice colour plates representing the symptoms and interactions on rachis are a very large contribution to the theme. I therefor do not have much major comment to do about the work which may be published with few modification.

However I would recommend some clarification about the protocol that remains no easy to understand in particular concerning the inoculation onto pseudosclerotial rachises.

-          L221. Add a reference to table 1 to be completely clear about isolates tested.

-          L228-230. Is that only one plate per interaction or do you have replicates.

-          L249-265. These 2 paragraphs are not clear and would benefit some rewriting. Describing the in vitro test first would make the paragraph clearer. For the in vivo test, 5 petioles with >20 apothecia in a limited area. I suppose this is the area recovered with conifer densely needled shoots from L264. Give the approximate area (range). This represent one interaction rachis/putative interacting fungus? Is there just one of those set-up per tested interaction? This is done for the 36 species of table 1 I presume? I understood that the inoculation was by putting a colonised agar plug on the 2 end of each rachis, both in vitro and in vivo. Is that correct? For the in vitro test (L257): how many rachises per interaction? 3= one per glass rod? This would make (3+5)*36= 288 which is in the range of the 250 per year given in L250 but not equal, so likely I did not get everything. L277-279: This refers to the in vitro set-up only? For the in vivo one, same or just an end observation?

-          Discussion L607-609. Also, although the experiment was done on a substantial period, it remains less than a month. The impact on a vegetative season may be much more. You may also indicate that H. fraxineus persist several year on the rachis, which increases the overall amount of litter inoculum. Whatever may reduce this persistence may be very significant.

Author Response

We would like to thank you for the time and effort given to reviewing our manuscript and for the comments and suggestions. All your suggestions  have been taken into account.

Question 1 L221. Add a reference to table 1 to be completely clear about isolates tested.

Answer 1: It is done.

Question 2 L228-230. Is that only one plate per interaction or do you have replicates.

Answer 2 : Information is added in line 226

It was one plate per interaction, but three strains of H. fraxineus were included. In fact if there were any disruptions [e.g. no growth of H. fraxineus (growth was sometimes concentrated only on inoculum) or easy dispersal of spores and development of multiple colonies – in case of Cordyceps sp.] there were new multiple repeats. However, we treat it as a technical problem.

Question 3 - L249-265. These 2 paragraphs are not clear and would benefit some rewriting. Describing the in vitro test first would make the paragraph clearer. For the in vivo test, 5 petioles with >20 apothecia in a limited area. I suppose this is the area recovered with conifer densely needled shoots from L264. Give the approximate area (range). This represent one interaction rachis/putative interacting fungus? Is there just one of those set-up per tested interaction? This is done for the 36 species of table 1 I presume? I understood that the inoculation was by putting a colonised agar plug on the 2 end of each rachis, both in vitro and in vivo. Is that correct? For the in vitro test (L257): how many rachises per interaction? 3= one per glass rod? This would make (3+5)*36= 288 which is in the range of the 250 per year given in L250 but not equal, so likely I did not get everything. L277-279: This refers to the in vitro set-up only? For the in vivo one, same or just an end observation?

Answer 3

Changes have been made to the text as suggested, numerous details have been provided. We hope everything is clear now. About 250 petioles were collected each year, much more than the amount used in the experiment (see Supplementary Table 1). L277-279: Text is for both types of tests. In vivo, still other technical activities were performed, which is not mentioned in the manuscript: e.g. every 2-3 days it was necessary to collect and remove various types of snails from places where petioles were laid out to another area (it has long been observed that snails eagerly eat pseudosclerotia, as well as apothecia).

Question 4 Discussion L607-609. Also, although the experiment was done on a substantial period, it remains less than a month. The impact on a vegetative season may be much more. You may also indicate that H. fraxineus persist several year on the rachis, which increases the overall amount of litter inoculum. Whatever may reduce this persistence may be very significant.

Answer 4: The text has been supplemented in line with the above comments.

Reviewer 2 Report

I was impressed by the thoroughness of this study, both as concerns the huge amount of work done in the lab and in the forest, as well as the meticulous documentation of the interactions between the fungi and H. fraxineus. The paper was well written and easy to read, and I have only a few corrections and comments, which are included in the pdf file.

In addition, I have only two comments:
It would have been interesting if you had been able to obtain mycelium (or maybe even fruitbodies) of Hymenoscyphus albidus (the native endophyte on ash leaves) and tested the fungi against it to see whether there was any difference. However, I  appreciate that this would have been quite difficult, since the fungus has been outcompeted in most of Europe.

My other comment is about the potential use of these fungi as biocontrol agents. This is not really discussed in the paper, and I would have liked to see at least a small paragraph to increase the relevance of the study. In my opinion, the main use for BCAs against H. fraxineus would not be in forest situations, but rather in urban settings and the open landscape to protect valuable individuals. We already observe that there is less impact of ash dieback outside forests, and one of the possible explanations is that the infection pressure is much lower in the drier and windier conditions which often prevail outside forests. Even though ascospores of H. fraxineus have been shown capable of travelling many kilometers, it could still be assumed that inoculum from local sources are most important for infection. Especially, when it comes to infection via bark at the base of stem, which often opens the way for subsequent attacks by honey fungus (Armillaria sp.). Preventing such infections, eg by applying the BCAs on stem bases and the near vicinity, could be very useful. Protecting prominent ash trees in urban settings and the open landscape would be much more likely, that trying to use BCAs in forests, where replanting with tolerant ash, based on genetic selection, seems a better long-term strategy.
I would appreciate knowing your thoughts on this aspect, but it need not be a long discussion or firm conclusions, as I agree that more research and experiments are probably needed.

Author Response

We would like to thank you for the time and effort given to reviewing our manuscript and for the comments and suggestions included in the pdf file. All suggestions have been implemented in the text. Only in line 460, instead of the proposed 'fresh petioles', we wrote 'symptomless or dead petioles'. 

Question 1  It would have been interesting if you had been able to obtain mycelium (or maybe even fruitbodies) of Hymenoscyphus albidus (the native endophyte on ash leaves) and tested the fungi against it to see whether there was any difference. However, I  appreciate that this would have been quite difficult, since the fungus has been outcompeted in most of Europe.

Answer 1

Despite great efforts and searches, in Poland, since the occurrence of ash dieback, we have not been able to isolate the mycelium or find the apothecia of Hymenoscyphus albidus. Some of the fungal taxa found now are rather not mycohost specific and would probably be able to attack also H. albidus. For example, from the dying apothecia of Hymenoscyphus pusillus on F. pennsylvanica petioles in South Poland, Clonostachys rosea, as in case of H. fraxineus, was isolated. If H. albidus were available, it should be borne in mind that carrying out additional experiments with this species would be a major undertaking.

Question 2  

My other comment is about the potential use of these fungi as biocontrol agents. This is not really discussed in the paper, and I would have liked to see at least a small paragraph to increase the relevance of the study

Answer 2

The last paragraph has been extended to include these items. We believe that where possible (e.g. urban settings, open landscape), raking leaves and the procedure shown in Noble et al. 2019 (paper cited under number 13) should be preferred. We believe that the use of biocontrol fungi would be possible in the forest and in larger groups of trees in urban green areas to reduce the inoculum reservoir of H. fraxineus in the litter. Causing mass colonization of petioles in the litter e.g. by some Basidiomycota could cause freshly fallen leaves to be further colonized in the following years and there would be a long-term reduction in inoculum in the litter. Another possibility is, as you suggest, preventing of infections at the base of stems.